# Inhibition of SRC-3 as a potential therapeutic strategy for aggressive mantle cell lymphoma

Imani Bijou[1], Yang Liu[2], Dong Lu[1], Jianwei Chen[1], Shelby Sloan[3], Lapo Alinari[3], David M. Lonard[4]*, Bert W. O'Malley[4]*, Michael Wang[2]*, Jin Wang[1,4]*

1 Department of Biochemistry and Molecular Pharmacology, Baylor College of Medicine, Houston, Texas, United States of America, 2 Department of Lymphoma and Myeloma, The University of Texas MD Anderson Cancer Center, Houston, Texas, United States of America, 3 Division of Hematology, The James Cancer Hospital and Solove Research Institute, The Ohio State University, Columbus, Ohio, United States of America, 4 Department of Molecular and Cellular Biology, Baylor College of Medicine, Houston, Texas, United States of America

* wangj@bcm.edu (JW); miwang@mdanderson.org (MW); berto@bcm.edu (BWO); dlonard@bcm.edu (DML)

**Data Availability Statement:** All relevant data are within the manuscript and its Supporting information files.

**Funding:** The research was supported in part by National Institute of Health (R01-CA207701 and

## Abstract

Mantle cell lymphoma (MCL) has a poor prognosis and high relapse rates despite current therapies, necessitating novel treatment regimens. Inhibition of SRC-3 show effectiveness *in vivo* and *in vitro* in other B cell lymphomas. Additionally, previous studies have shown that SRC-3 is highly expressed in the lymph nodes of B cell non-Hodgkin's lymphoma patients, suggesting SRC-3 may play a role in the progression of B cell lymphoma. This study aimed to investigate novel SRC-3 inhibitors, SI-10 and SI-12, in mantle cell lymphoma. The cytotoxic effects of SI-10 and SI-12 were evaluated *in vitro* and demonstrated dose-dependent cytotoxicity in a panel of MCL cell lines. The in vivo efficacy of SI-10 was confirmed in two ibrutinib-resistant models: an immunocompetent disseminated A20 mouse model of B-cell lymphoma and a human PDX model of MCL. Notably, SI-10 treatment also resulted in a significant extension of survival in vivo with low toxicity in both ibrutinib-resistant murine models. We have investigated SI-10 as a novel anti-lymphoma compound via the inhibition of SRC-3 activity. These findings indicate that targeting SRC-3 should be investigated in combination with current clinical therapeutics as a novel strategy to expand the therapeutic index and to improve lymphoma outcomes.

## Introduction

Non-Hodgkin lymphoma (NHL) is the most commonly occurring hematological malignancy containing a variety of subtypes. Mantle cell lymphoma (MCL) is an aggressive and incurable subtype of B-cell-NHL, accounting for 4% of all lymphomas and resulting in a median survival of 8–12 years [1,2]. Most patients present with advanced-stage disease, and current therapies, which include anti-CD20 monoclonal antibodies, autologous stem-cell transplantation, immunochemotherapy, and targeted therapy, have been unable to eradicate MCL resulting in almost universal relapse [3]. Given the current efficacy of standard lymphoma treatment regimens, therapies have been investigated targeting B-cell receptor pathway signaling or apoptotic

R01-268518 to J.W., R01-CA250503 to J.W. and M.W.), Cancer Prevention & Research Institute of Texas (CPRIT, RP170500 to B.W.O. and J.W.), and the Michael E. DeBakey, M.D., Professorship in Pharmacology (to J.W.). The funder provided support in the form of salaries for authors [DML, BWO], but did not have any additional role in the study design, data collection and analysis, decision to publish, or preparation of the manuscript. The specific roles of these authors are articulated in the 'author contributions' section.

**Competing interests:** J.W. is the co-founder of CoActigon Inc. and Chemical Biology Probes LLC. D.M.L. and B.W.O. are co-founders of CoRegen Inc. J.W. serves as a consultant for CoRegen Inc. Although with commercial affiliations, this does not alter our adherence to PLOS ONE policies on sharing data and materials.

pathway signaling to treat relapsed/refractory disease [1]. The BTK inhibitor ibrutinib and some of its analogs have resulted in high overall response rates and FDA approval. However, resistance and relapse still occur resulting in treatment regimens combining other targeted therapies and necessitating approaches with differing mechanisms of action to those clinically approved [4–6].

The steroid receptor coactivator (SRC) family contains three members, SRC-1, SRC-2, and SRC-3 [7]. These transcriptional coactivators function through interactions with nuclear receptors which then recruit additional proteins to form multi-subunit transcriptional complexes that promote transcriptional activity. Additionally, SRCs have been shown to coactivate various non-nuclear receptor transcriptional factors such as NF-kB, AP-1, and E2F1, and interact with the CBP/p300 coactivators [4]. The dysregulation of epigenetic modulators is an initiator of carcinogenesis, and SRC-3 overexpression has been associated with malignancy in breast, lung, and prostate cancer [8–10]. SRC-3 has also been shown to promote tumor growth through involvement in pathways regulating cell cycle, apoptosis, drug resistance, migration, and invasion [11].

The role of SRC-3 in MCL yet to be well defined, but previous studies show that SRC-3 is highly expressed in B-cell NHL models. Pharmacological inhibition of SRC-3 with gambogic acid reduced tumor growth of diffuse large cell b cell lymphoma (DLBCL) models *in vitro* and *in vivo* through histone deacetylation and downregulation of multiple oncoproteins such as Bcl-2, cyclin D3, Bcl-6, and c-Myc [12]. Additionally, single-cell transcriptomic data suggest that SRC-3 is most highly expressed in B cells and plasma cells, suggesting a potential undiscovered role of SRC-3 in B cell receptor signaling [13].

In collaboration with Drs. O'Malley and Lonard's groups, we identified SI-2 as a potent SRC-3 inhibitor capable of decreasing SRC-3 protein levels and inhibiting breast cancer cell proliferation *in vitro* and tumor growth *in vivo* [14]. More recently, SI-2 was optimized for more favorable pharmacokinetic properties by introducing of up to three fluorine atoms generating the analogs SI-10 and SI-12. These analogs have been shown to exert potent anti-cancer activity with minimal cardiotoxicity in breast cancer models [15]. In this study, we aim to investigate the anti-cancer effect of new SRC-3 inhibitors *in vitro* and *in vivo* for the treatment of MCL.

## Results

### Src-3 inhibitors reduce the proliferation of various MCL cell lines

Previous studies have shown SRC-3 overexpression to be pivotal in many solid tumors, but the significance of SRC-3 in blood cancer tumors was undetermined [8–10]. Recently, a study showed clinical SRC-3 overexpression in the lymph nodes of B-cell NHL patients [12]. To evaluate the in vitro anti-lymphoma activity of SI-10 and SI-12, four MCL lines and one murine lymphoma line were treated with either SRC-3 inhibitor SI-10 or SI-12 at concentrations ranging from 0–2 μM. Cell viability was assessed using the Alamar Blue assay after 48 hours. After treatment, both SI-10 and SI-12 significantly inhibited the growth of lymphoma in a dose-dependent manner. $IC_{50}$ values were calculated at the low nanomolar level for both SI-10 and SI-12 (Fig 1A and 1B). Comparing the $IC_{50}$ values of the two drugs, they both demonstrated similar efficacy in the human MCL cell lines. The murine A20 B cell lymphoma line showed the most resistance to SI-10 with an $IC_{50}$ of 20 nM. Cell viability was also measured at different treatment times 12, 24, and 48 h at concentrations up to 2 μM. Viability also decreased in a time-dependent manner (Fig 1C and 1D).

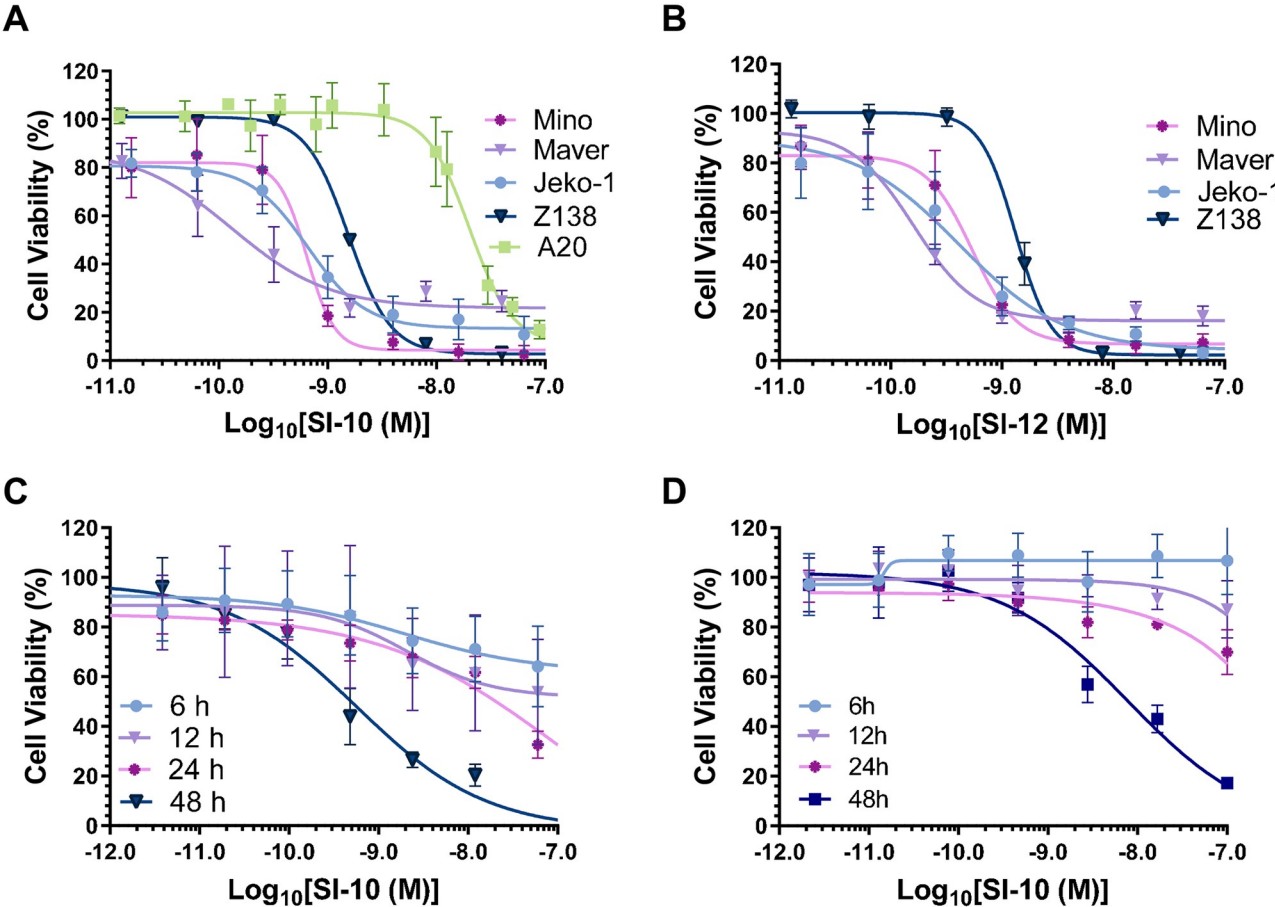

**Fig 1. MCL cell lines were treated with SRC-3 inhibitors.** A) SI-10 or B) SI-12 at various concentrations for 48 h and cell viability was determined using the resazurin assay C) Mino D) Jeko-1 growth inhibition with SI-10 (0–1 mM) for 6,12,24, or 48h.

### SRC-3 inhibitors overcome drug resistance *in vitro*

The first-line treatment of MCL includes chemotherapy and the anti-CD20 monoclonal antibody rituximab, but given that so many patients relapse, targeted therapies are frequently offered as a second-line option. Current targeted therapy for MCL includes the FDA-approved BTK inhibitor ibrutinib and the Bcl-2 inhibitor venetoclax currently in clinical trials. B–cell receptor (BCR) signaling is highly upregulated in B-cell malignancies, and inhibition of BTK leads to durable clinical responses in MCL [16]. Additionally, Bcl-2 regulates the intrinsic mitochondrial apoptotic pathway where its overexpression results in mitochondrial outer membrane permeabilization through the interplay of pro- and anti-apoptotic Bcl-2 family proteins [17]. Despite the clinical efficacy of both drugs, resistance to venetoclax and ibrutinib is still of concern as most patients progress on both drugs [18,19].

High levels of SRC-3 have been associated with resistance to chemotherapy and targeted therapies in cancer. SRC-3 has been associated with resistance to chemotherapy and targeted therapies in cancer models. Targeting SRC-3 with bufalin resulted in reduced polarization of pro-tumorigenic M2 macrophages by decreasing MIF expression in chemo-resistant colon cancer models [20]. Additionally, siRNA downregulation of SRC-3 reverses tamoxifen resistance in endocrine-resistant, HER2-positive breast cancer cells [21]. We next investigated if

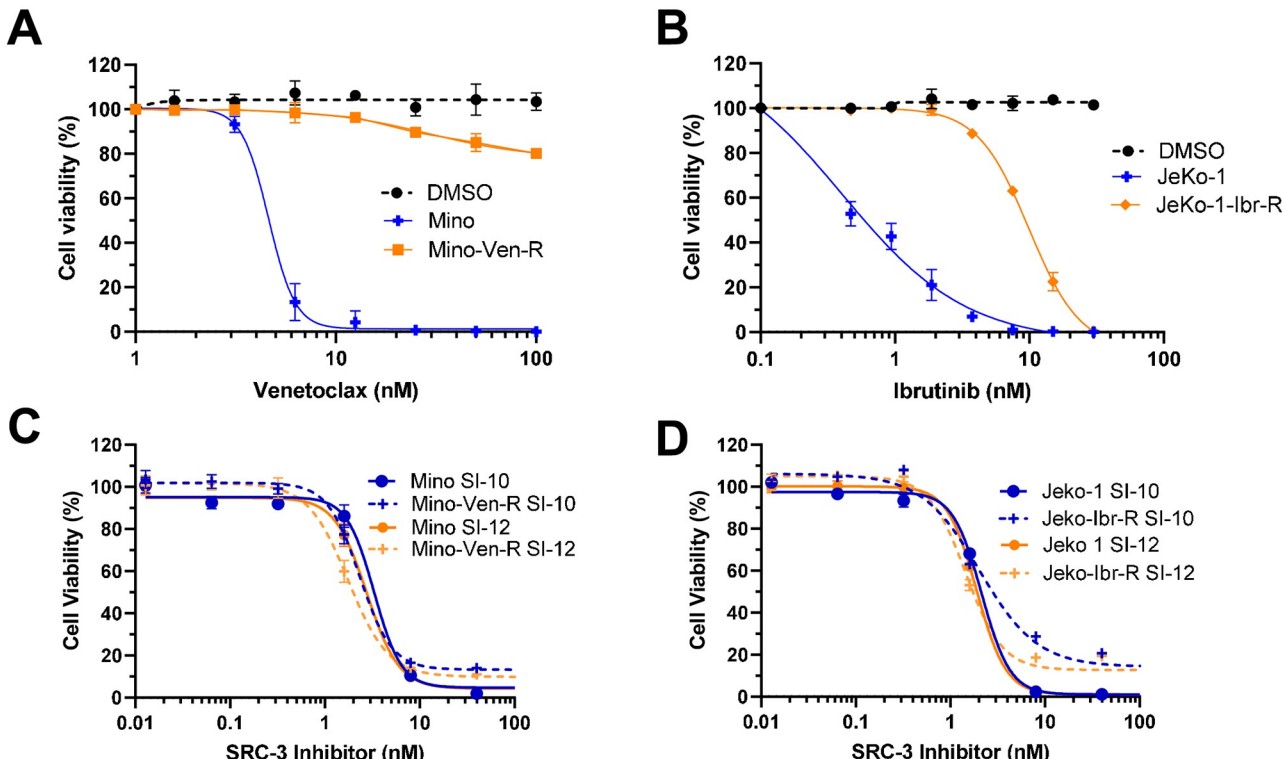

**Fig 2. Mino parental and Mino venetoclax resistant cells were treated with various concentrations of venetoclax.** B) Jeko-1 parental and Jeko-1 ibrutinib resistant cells were treated with ibrutinib at various concentrations. C) Mino parental and resistant cells were treated with SI-10 and SI-12. D) Jeko-1 parental and resistant cells were treated with SI-10 and SI-12.

SRC-3 inhibitors were effective in drug-resistant Jeko-1 and Mino cells given their efficacy in the parental cell lines. Venetoclax and ibrutinib-resistant mantle cell lymphoma lines were established in Dr. Michael Wang's lab by exposing the cells to stepwise dose increases of the drug [22,23]. Compared to the parental cells, both venetoclax and ibrutinib-resistant cells showed reduced sensitivity when treated with their respective inhibitors (Fig 2A and 2B). The efficacy of both SRC-3 inhibitors, SI-10 and SI-12, was then evaluated in the resistant lines. The SRC-3 inhibitors maintained nanomolar $IC_{50}$ in venetoclax-resistant Mino cells and ibrutinib-resistant Jeko-1 cells (Fig 2C and 2D). This suggests that SRC-3 inhibitors may be effective in drug-resistant models of MCL.

## SI-10 overcomes ibrutinib resistance in pdx mouse models

Most MCL patients eventually relapse; therefore it is critical to find drugs that are effective even after primary treatment. The MCL lines screened had varying sensitivities to ibrutinib, but all had low nanomolar $IC_{50}$ for our SRC-3 inhibitors. To evaluate the efficacy of SI-10 against ibrutinib resistance, we established a PDX mouse model with ibrutinib resistance in 6-week-old NSG mice by subcutaneous injection of ibrutinib-resistant Jeko cells. Mice were treated for 7 weeks five times a week with either vehicle, ibrutinib (50 mg/kg), or SI-10 (50 mg/kg) via oral gavage. Here we see reduced tumor volume and tumor weight after treatment between SI-10 and vehicle, and we also see an improvement compared to ibrutinib treatment (Fig 3A and 3B). Additionally, no significant body weight changes were observed throughout

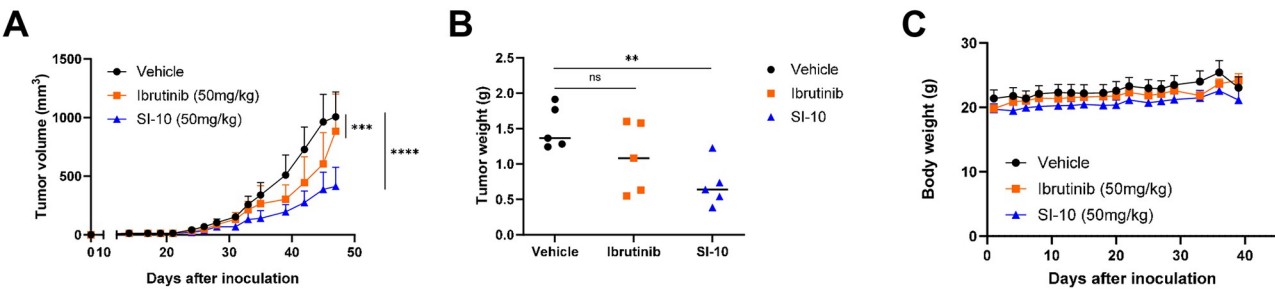

**Fig 3. SRC-3 treatment decreased tumor growth in a Jeko-1 ibrutinib resistant mouse model.** A) Volumes of tumors treated with SI-10 (50 mg/kg), ibrutinib (100mg/kg) or vehicle B) Weight of tumors treated with SI-10 (50 mg/kg), ibrutinib (100mg/kg) or vehicle before sacrificing C) The weight of each mouse was measured 3 times a week.

the treatment course (Fig 3C). This suggests that SI-10 could be beneficial against ibrutinib-resistant MCL.

## SI-10 inhibition extends survival in a syngeneic tumor model

We next investigated SI-10 in an A20 immunocompetent syngeneic mouse model of B cell lymphoma. With increasing knowledge of the tumor microenvironment, preclinical studies show that the TME plays a significant role in drug resistance to current therapies [24]. The A20 model exhibits clinical characteristics of MCL in vivo, including infiltration of the bone marrow and liver, and enlargement of the spleen [25]. This model has also been shown to exploit evasive immune mechanisms across tumor progression. Specifically, injection of A20 cells has been shown to induce an expansion of regulatory T cells [26]. Tumors generate an immunosuppressive environment to maintain optimal growth, and recent studies show SRC-3 may contribute to this suppressive environment. Inhibition of SRC-3 may act dually, reducing cancer cell growth and blocking the activity of suppressive immune cells like T regulatory cells (Tregs); this has been shown in vivo in breast cancer models [27,28]. We evaluated if SRC-3 inhibition would still be effective in an immunosuppressive environment. A20 cells were injected via the tail vein of Balb/c mice. A week later, mice were randomized into two groups (n = 7) to receive SI-10 at 1 mg/kg vehicle. Mice received daily intraperitoneal injections over 7 weeks with SI-10 or vehicle. During the treatment period, mice were evaluated for gross abnormalities associated with the model and sacrificed according to protocol. As shown, mice treated with 1 mg/kg of SI-10 have a median survival of 47 days, significantly extending survival compared to the vehicle-treated mice with a median survival of 33 days (Fig 4A). No noticeable side effects were observed during treatment and body weight was not significantly affected (Fig 4B).

## Chronic treatment with SI-10 is well tolerated

To evaluate chronic toxicity, ICR mice were treated daily with 50 mg/kg or 100 mg/kg SI-10 for 4 weeks and serum chemistry analysis was performed. The highest dose is 2 times that used in the mouse xenograft model above. Throughout the treatment course, we see no significant body weight fluctuations (Fig 5A). Compared to literature reference values and the control, no obvious difference was observed in the levels of liver enzymes AST and ALT (Fig 5B). Only one mouse in the 100 mg/kg group had high AST levels. Regarding kidney function, BUN levels were observed to be comparable to that of the reference range, although a bit lower than the control.

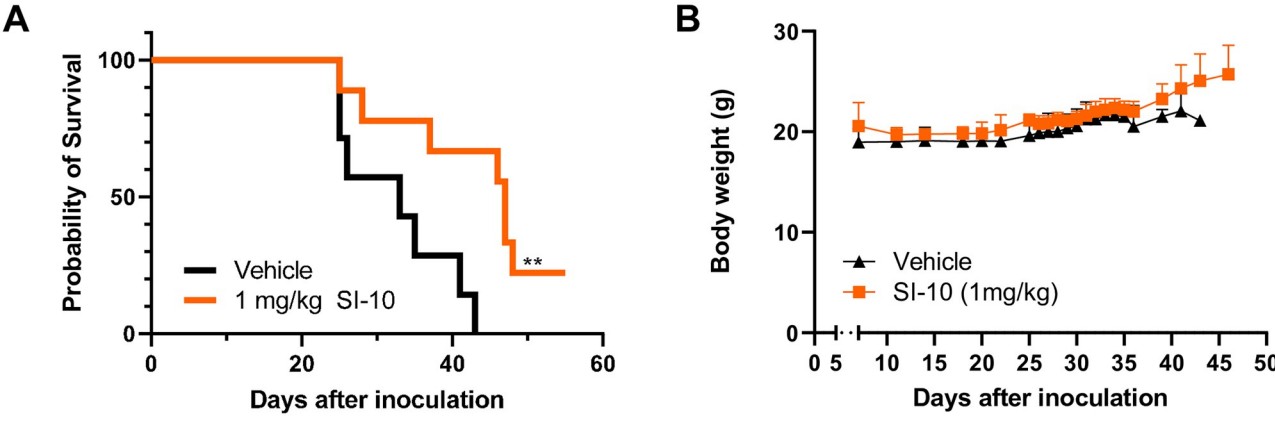

**Fig 4. SRC-3 treatment prolonged survival of A20-Balb/c mice.** Balb/c mice were injected with 1 x $10^6$ A20 cells intravenously. Treatment started on day 7. A) Kaplan-Meier survival plot reflecting time to lethal tumor burden. Based on the log-rank test, there are significant differences between the treated group and the control (P < 0.05). B) Body weight of all groups were measured 5 days a week.

## Discussion

Much research has been done exploring the role of SRC-3 in carcinogenesis through the regulation of oncoproteins and transcription factors in solid tumors, but the role of SRC-3 in blood cancers is less clear. SRC-3 is overexpressed in the lymph nodes of other B cell malignancies, but its role in MCL is not well characterized. MCL patients frequently undergo relapse to current therapies, necessitating novel treatment modalities to improve outcomes. This study

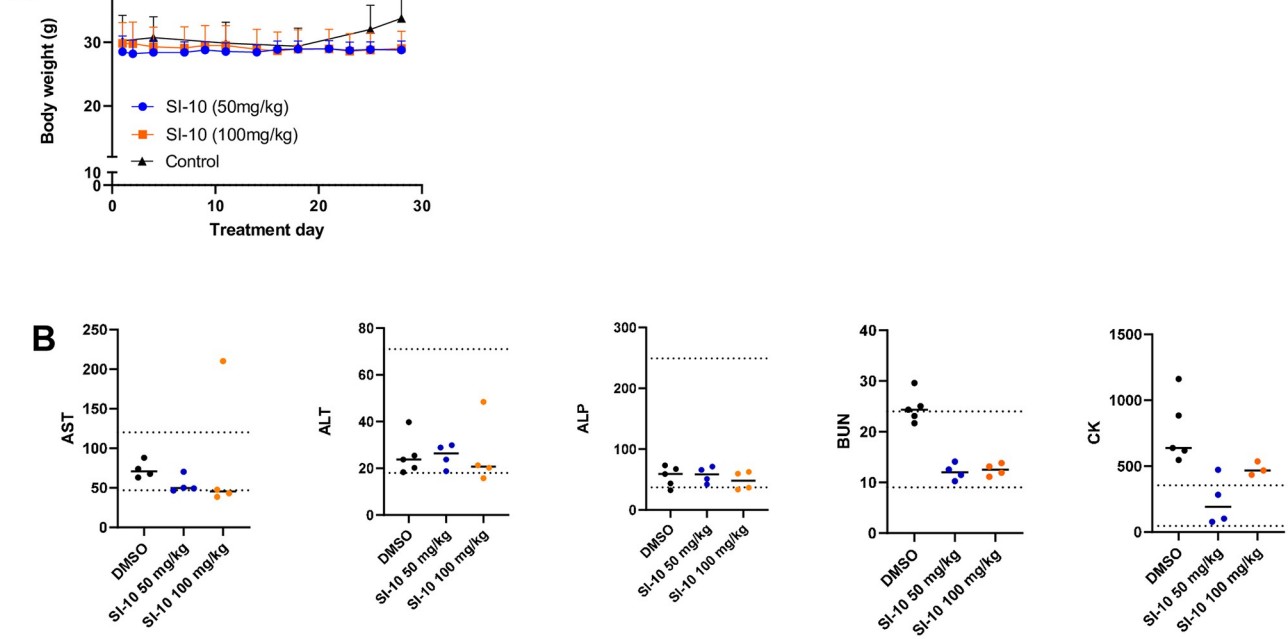

**Fig 5. SRC-3 inhibitor treatment in normal mice show little to no toxicity after treatment for 28 days.** A) Body weight of all groups were measured 5 days a week. B) Blood serum levels of clinical markers related to kidney and liver failure. Dotted lines represent upper and lower limits based on reference values. Data outliers were removed using the identify outlier function in Graphpad Prism with the ROUT method (Q = 1%).

demonstrates the therapeutic potential of SRC-3 inhibitor SI-10 as a treatment for *in vitro* and *in vivo* models of mantle cell lymphoma. SI-10 also exhibited inhibitory effects in drug-resistant models of MCL.

Previous data surrounding SRC-3 and B-cell malignancies is a bit conflicting regarding the possibility of SRC-3 as a tumor suppressor or oncogene. SRC-3 overexpression has been shown to contribute to other aggressive non-Hodgkin B cell lymphomas. Inhibition of SRC-3 in DLBCL with gambogic acid resulted in cell cycle arrest and apoptosis in B cell NHL lines. Treatment was also associated with the downregulation of DLBCL oncoproteins, including Bcl-2, Bcl-6, c-Myc, and NF-kB [12]. Unlike gambogic acid, SRC-3 has been confirmed to be a target of SI-10 [15]. Gambogic acid has been shown to target multiple cancer-related proteins in addition to SRC-3, including Bcl-2 family proteins, the proteasome, and topoisomerase IIa exhibiting polypharmacology [29–31]. Despite efficacy with pharmacological inhibition of SRC-3, complete amelioration of SRC-3 via knockout *in vivo* shows the specific induction of extreme lymphoproliferation of both T and B cells, eventually progressing into B-cell lymphoma with age. Interestingly, the *in vivo* studies showed no effect on cell proliferation and apoptosis in other tissue types in the SRC-3 knockout mice [32].

We demonstrate here that SI-10 is a potent anti-tumor small molecule for the treatment of MCL *in vitro* and *in vivo*. SI-10 exhibits low nanomolar efficacy in a panel of MCL cell lines. SRCs have been shown to regulate a multitude of pathways involved in cancer progression and metastasis, including known targets of MCL. Several signaling molecules have been implicated in MCL pathogenesis, including PI3K/AKT, NF-kB, and Bcl-2 [33]. SRC-3 has previously been shown to coactivate many of the targets important in MCL pathogenesis in other cancer types [10,34,35]. Many of these same pathways are also implicated in ibrutinib resistance. Ibrutinib-resistant cells activate BCR signaling through the PI3K/AKT pathway and NF-kB signaling, and maintain cell cycle progression through cyclin D1 to undergo primary resistance [19]. These findings suggest that SRC-3 signaling may be important for MCL cell survival and drug resistance through the possible coactivation of these targets.

In addition to its efficacy *in vitro*, SI-10 has demonstrated survival extension in an immunocompetent lymphoma model and a human PDX model both resistant to ibrutinib. The role of SRC-3 in cancer drug resistance is context-dependent, but previously SRC-3 overexpression has been shown to contribute to Herceptin resistance in ERBB2 overexpressing breast cancer cells [36]. Given the high rate of relapse against current MCL treatments, overcoming drug resistance is especially exciting and warrants further study. SI-10 also has a good safety profile. In the studies performed here, mice in both treated and control groups exhibited normal behavior and minimal body weight loss. SI-10 treatment was well tolerated even at 50 times the dose tested for survival extension.

In conclusion, SI-10 is a small molecule SRC-3 inhibitor that can inhibit MCL *in vitro* and *in vivo* and overcome ibrutinib resistance. With further studies, SI-10 may be a promising therapeutic candidate for ibrutinib-resistant MCL.

## Methods

### Cell culture

Cells were grown in RPMI-1640 supplemented with 10% fetal bovine serum (FBS) and 1% penicillin-streptomycin (10000 U/mL) and cultured at 37°C in a humidified incubator at 5% $CO_2$ for all experiments. Human MCL cell lines (JeKo-1, Mino, Maver-1, Z-138, Jeko-1-IbrR, Mino-VenR) were gifted from Dr. Michael Wang's lab (MD Anderson). The murine-derived A20 cells are from ATCC and were maintained in RPMI-1640 supplemented with 0.05 mM 2-mercaptoethanol, 10% FBS, and 1% penicillin-streptomycin.

## Cell viability assay

Cells were seeded in 96 well plates at a density of 5000–8000 cells per well. The next day cells were treated with serially diluted SRC-3 inhibitors for 48 h along with DMSO control. Cell viability was measured using the Alamar Blue assay. Resazurin was added at 10% of well volume and incubated with cells for 4 hours. Fluorescence was measured at excitation/emission 544/590 nm. Viability was calculated by plotting viability relative to control. The $IC_{50}$ values for compounds were calculated based on the Hill-Slope equation and analyzed in GraphPad Prism 9 (GraphPad Software, Inc., San Diego, CA, USA).

## In vivo experiments

**Therapeutic efficacy of SI-10 in ibrutinib-resistant MCL PDX mice.** To establish a xenograft model, 5 million cells were injected subcutaneously into NSG mice (6–8 weeks old). After the formation of palpable tumors, mice were treated orally 5 days per week with SI-10 50 mg/kg, ibrutinib 50 mg/kg, or vehicle (DMSO) (n = 5 for all groups) for 7 weeks. Tumor size and body weight were measured 3 times a week. Animals were monitored five times a week and tumor volume was measured using callipers every other 2–3 days in three dimensions. Mice were euthanized either when the tumor diameter reached the protocol limit of 1.5 cm or if the tumor showed signs of ulceration reaching 4mm.

**Survival extension of SI-10 in A20 lymphoma mice.** To evaluate the survival extension of SI-10 in vivo, immunocompetent tumor xenograft models were developed in female Balb/c mice (6–8 weeks old, Jackson Labs) via the tail vein injection of $1 \times 10^6$ B cell lymphoma cells. (A20). A week later, mice were randomized using Graphpad and treated with 1 mg/kg/day SI-10 (n = 9) or vehicle (DMSO) (n = 7) control via intraperitoneal injection for 8 weeks. Animals were monitored daily and sacrificed based on the experimental protocol.

**In vivo toxicity.** Female ICR mice (6–8 weeks old, CCM Vendor) were treated with 50 mg/kg/day SI-10 (n = 5) or 100 mg/kg/day SI-10 (n = 5) daily by oral gavage for 4 weeks. Control mice (n = 5) were treated with the vehicle (DMSO). After the treatment period, 200 μL of blood was collected from the mice via submandibular bleeding to isolate plasma for conducting a comprehensive panel of serum chemistry assays. Animals were monitored daily for signs of toxicity.

As per the experimental procedure approved by the Institutional Animal Care and Use Committee at BCM and MD Anderson Cancer Center, mice were monitored at least every 3 days to ensure no more than a 10% decrease in body weight. Animal welfare considerations were taken to minimize suffering and distress. Daily monitoring was required if the 10% weight loss threshold was met or once tumor is over 1.0 cm in diameter. Additionally, any mice displaying signs of distress, including immobility, huddled posture, inability to eat, ruffled fur, self-mutilation, vocalization, wound dehiscence, hypothermia, or a weight loss greater than 20%, were humanely euthanized using isoflurane the same day following guidelines. No animals died before meeting the criteria for euthanasia. Imani Bijou completed training from the Center for Comparative Medicine (CCM) at BCM.

## Statistical analysis

All statistical graphs are constructed using Prism 9 (GraphPad Software, Inc.). $P < 0.05$ was considered statistically significant.

## Supporting information

**S1 File.**
(XLSX)

## Author Contributions

**Conceptualization:** Imani Bijou, David M. Lonard, Bert W. O'Malley, Michael Wang, Jin Wang.

**Data curation:** Imani Bijou, Yang Liu, Dong Lu, Jianwei Chen, Shelby Sloan.

**Formal analysis:** David M. Lonard.

**Funding acquisition:** David M. Lonard, Bert W. O'Malley, Michael Wang, Jin Wang.

**Investigation:** Imani Bijou.

**Methodology:** Imani Bijou.

**Project administration:** Michael Wang.

**Resources:** Lapo Alinari.

**Supervision:** Bert W. O'Malley, Jin Wang.

**Writing – original draft:** Imani Bijou, David M. Lonard, Bert W. O'Malley, Jin Wang.

**Writing – review & editing:** Yang Liu, Dong Lu, Jianwei Chen, Shelby Sloan, Lapo Alinari, David M. Lonard, Bert W. O'Malley, Michael Wang, Jin Wang.

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
