## [Decision Letter · Decision Letter 0]

15 Dec 2023

PONE-D-23-23808

Inhibition of SRC-3 as a potential therapeutic strategy for aggressive mantle cell lymphoma

PLOS ONE

Dear Dr. Wang,

Thank you for submitting your manuscript to PLOS ONE. After careful consideration, we feel that it has merit but does not fully meet PLOS ONE’s publication criteria as it currently stands. Therefore, we invite you to submit a revised version of the manuscript that addresses the points raised during the review process.

We look forward to receiving your revised manuscript.

Kind regards,

Francesco Bertolini, MD, PhD

Academic Editor

PLOS ONE

Journal Requirements:

   "The research was supported in part by National Institute of Health (R01-CA207701 to J.W., R01-CA250503 to J.W. and M.W.), Cancer Prevention & Research Institute of Texas (CPRIT, RP170500 to B.W.O. and J.W.), and the Michael E. DeBakey, M.D., Professorship in Pharmacology (to J.W.)."

   "The research was supported in part by National Institute of Health (R01-CA207701 to J.W., R01-CA250503 to J.W. and M.W.), Cancer Prevention & Research Institute of Texas (CPRIT, RP170500 to B.W.O. and J.W.), and the Michael E. DeBakey, M.D., Professorship in Pharmacology (to J.W.)."

   "The research was supported in part by National Institute of Health (R01-CA207701 to J.W., R01-CA250503 to J.W. and M.W.), Cancer Prevention & Research Institute of Texas (CPRIT, RP170500 to B.W.O. and J.W.), and the Michael E. DeBakey, M.D., Professorship in Pharmacology (to J.W.)."

   "J.W. is the co-founder of CoActigon Inc. and Chemical Biology Probes LLC. D.M.L. and B.W.O. are co-founders of CoRegen Inc. J.W. serves as a consultant for CoRegen Inc."    

We note that one or more of the authors are employed by a commercial company: CoActigon Inc. and Chemical Biology Probes,  CoRegen Inc.

7. Your ethics statement should only appear in the Methods section of your manuscript. If your ethics statement is written in any section besides the Methods, please move it to the Methods section and delete it from any other section. Please ensure that your ethics statement is included in your manuscript, as the ethics statement entered into the online submission form will not be published alongside your manuscript. 

8. PLOS ONE now requires that authors provide the original uncropped and unadjusted images underlying all blot or gel results reported in a submission’s figures or Supporting Information files. This policy and the journal’s other requirements for blot/gel reporting and figure preparation are described in detail at https://journals.plos.org/plosone/s/figures#loc-blot-and-gel-reporting-requirements and https://journals.plos.org/plosone/s/figures#loc-preparing-figures-from-image-files. When you submit your revised manuscript, please ensure that your figures adhere fully to these guidelines and provide the original underlying images for all blot or gel data reported in your submission. See the following link for instructions on providing the original image data: https://journals.plos.org/plosone/s/figures#loc-original-images-for-blots-and-gels. 

Reviewers' comments:

Reviewer's Responses to Questions

**Comments to the Author**

1. Is the manuscript technically sound, and do the data support the conclusions?

Reviewer #1: Yes

Reviewer #2: Yes

2. Has the statistical analysis been performed appropriately and rigorously? 

Reviewer #1: Yes

Reviewer #2: Yes

3. Have the authors made all data underlying the findings in their manuscript fully available?

Reviewer #1: Yes

Reviewer #2: Yes

4. Is the manuscript presented in an intelligible fashion and written in standard English?

Reviewer #1: Yes

Reviewer #2: Yes

5. Review Comments to the Author

Reviewer #1: MCL is malignant disease with poor clinical outcomes. In this study, the authors have shown inhibiting SRC-3 using SI-10 and SI-12 effectively inhibit MCL cells growth. Using in vivo approaches, the authors have shown inhibiting SRC-3 increase overall survival suggesting the potential of SRC-3 inhibitor in treating MCL. In this paper, proper approaches and statistical analysis have been applied.

Reviewer #2: Wang et al report preclinical data on an steroid receptor coactivator SRC inhibitor, named SI-10, in mantle cell lymphoma cell lines, especially in venetoclax or ibrutinib resistant cell lines and in a patient derived and syngeneic mouse model. +

A new drug for ibru or ven resistant mantle cell lymphoma is relevant application.

Question 1 cell model

The ibrutinib or venetoclex resistance is artificially produced. Do you know, or have hints if this resistance is similar to resistance seen in patients ? Did you test and/or find BTK mutations. ?

Question 2 long-term toxicty

Was there a concern for cardiotoxicity, as mentioned on page 2 ? You noted no weight changes in long-term treated mice.

Did you check for lab abnormalities troponin and pro-BNP, ?

6. PLOS authors have the option to publish the peer review history of their article (what does this mean?). If published, this will include your full peer review and any attached files.

Reviewer #1: No

Reviewer #2: No

---

## [Author Response · Author response to Decision Letter 0]

25 Feb 2024

Reviewers' comments:

Question 1 cell model

The ibrutinib or venetoclex resistance is artificially produced. Do you know, or have hints if this resistance is similar to resistance seen in patients ? Did you test and/or find BTK mutations. ?

We (The Michael Wang group) previously reported that the resistance mechanism in Mino-venetoclax-R cells is associated with increased AKT phosphorylation and decreased PTEN levels (Lan et al., Clinical Cancer Research, 24(16), 2018). Currently, we are in the process of delineating the resistance mechanism for JeKo-R. Our observations reveal an upregulated OXPHOS pathway in JeKo-R cells (unpublished data), aligning with the observed ibrutinib resistance mechanism in patient samples (Sci Transl Med. 2019 May 8;11(491):eaau1167). We have not compared the resistance of the cell line to patient samples.

BTK mutation is a common drug resistant mechanism in chronic lymphocytic leukemia (CLL). However, in mantle cell lymphoma (MCL), BTK mutation is not a major resistant mechanism (Zhao et al. Unification of de novo and acquired ibrutinib resistance in mantle cell lymphoma. Nat Commun. 2017;8:14920.). Therefore, we did not test BTK mutations in this study. Based on DepMap.org database, Mino, Maver1, and Jeko1 have wild type BTK.

Question 2 long-term toxicty

Was there a concern for cardiotoxicity, as mentioned on page 2 ? You noted no weight changes in long-term treated mice.

Did you check for lab abnormalities troponin and pro-BNP, ?

We did not check troponin and pro-BNP. But we did look at the creatine kinase levels and did not observe statistical elevation of CK levels upon SI-10 treatments. The data has been added to the manuscript in Figure 5.

---

## [Editor Report · Decision Letter 1]

10 Apr 2024

Inhibition of SRC-3 as a potential therapeutic strategy for aggressive mantle cell lymphoma

PONE-D-23-23808R1

Dear Dr. Wang,

We’re pleased to inform you that your manuscript has been judged scientifically suitable for publication and will be formally accepted for publication once it meets all outstanding technical requirements.

Kind regards,

Francesco Bertolini, MD, PhD

Academic Editor

PLOS ONE